# Rapid and green detection of manganese in electrolytes by ultraviolet-visible spectrometry to control pollutant discharge

Zhehua Xue[1,2], Lei Li[1,2]*

1 State Key Laboratory of Pollution Control and Resource Reuse, Tongji University, Shanghai, China,
2 Shanghai Institute of Pollution Control and Ecological Security, Shanghai, China

* lileitongji@126.com

**Data Availability Statement:** All relevant data are within the manuscript and its Supporting Information files.

## Abstract

Controlling the manganese ($Mn^{2+}$) concentration is important to product quality in the electrolytic manganese industry. Conventional methods for detecting $Mn^{2+}$ are complex and reagent-consuming, which makes them slow and polluting. It is of great significance to develop a new fast and environmentally friendly method to quantify $Mn^{2+}$ in electrolyte. The characteristic ultraviolet-visible (UV-vis) absorbance at 401 nm of $Mn^{2+}$ will vary linearly with the $Mn^{2+}$ concentration after data correction. Adjusting the pH, calibrating the spectral bandwidth (SBW) and optical path length (OPL), and subtracting the interference from coexisting substances by linear interpolation improve the measuring accuracy. $Mn^{2+}$ concentration can be determined by direct fast UV-vis absorbance measurement at characteristic peaks without using harmful reagents which facilitates such measurement to be applicated as on-line detection method for electrolytic manganese industry. The method developed in this study will help achieve the goal of improving the detection speed while cutting back on pollutant discharge from concentration analysis process in electrolytic manganese industry.

## Introduction

In electrolytic manganese industry, the concentration of manganese ($Mn^{2+}$), which is the major component in electrolytes, is of great concern. The $Mn^{2+}$ concentration can be neither too high nor too low during processes. In newly prepared purified leachate, the $Mn^{2+}$ concentration should be 34.0~38.0 g/L, while that in the electrolyzer cathode should be 15.0~18.0 g/L [1]. Exorbitant $Mn^{2+}$ concentrations will lead to operating malfunctions, such as excessively alkaline electrolytes, blackened products and manganese ammonium double-salt precipitation, whereas insufficient $Mn^{2+}$ will result in water electrolysis, plate flaking, and difficult stripping of the manganese plates [1]. An inappropriate manganese concentration leads to a higher rejection rate of electrolytic manganese products, and the risk of waste products being discharged and polluting the environment will consequently be greater. Heavy metals from industrial waste emissions, such as manganese, are well recognized as harmful. Hazards posed by manganese to flora and fauna have been extensively studied [2]. Chronic exposure to

**Funding:** This work was supported by the Fundamental Research Funds for the Central Universities (No grant number, received by Lei Li, URL: www.moe.gov.cn). The funder had no role in study design, data collection and analysis, decision to publish, or preparation of the manuscript.

**Competing interests:** The authors have declared that no competing interests exist.

manganese can lead to adverse neurological effects in humans [3, 4], including intellectual impairment in children [5]. Overall, it is necessary and important to develop a real-time manganese detection method for timely reporting of manganese concentration during the processing of industrial electrolytic manganese, which facilitates prompt adjustment of operating conditions to prevent malfunctions resulting from an abnormal concentration. When the malfunction rates reduce, the quantity of waste products from the electrolytic manganese industry will also decrease. This will directly save electricity and raw material consumption. More importantly, fewer waste products will indirectly reduce the risk of manganese pollution from industrial waste discharge.

To date, there have been many studies determining $Mn^{2+}$ concentration using different types of methods. Some researchers measured $Mn^{2+}$ concentration by colorimetry based on the chromogenic reaction between manganese ion and color-developing agents [6, 7]. The timeliness of this kind of methods is not good due to the relatively long time needed for the completion of chromogenic reaction. Such methods can only measure $Mn^{2+}$ at low concentrations. However, the concentrations of $Mn^{2+}$ in leaching solutions and electrolytes of electrolytic manganese industry are generally as high as dozens of grams per liter [1]. If colorimetry is used for $Mn^{2+}$ concentration determination in industry, laborious dilution will be needed before measurement. Moreover, the high cost of color-developing agents will limit the application of colorimetry in industry. Advanced fluorosensor for trace metal ion detection were synthesized in previous study [8]. But it is not suitable for determination of the high concentrations of $Mn^{2+}$ in industry. Titrimetric analysis is routinely the foundation of standard methods for measuring $Mn^{2+}$ concentrations [9, 10]. But the analysis speed of titrimetry is also slow. Besides, toxic reagents such as ammonium sulfide, eriochrome black T, EDTA, N-phenylanthranilic acid, and ammonium iron (II) sulfate are indispensable for $Mn^{2+}$ concentration measurement by titration. These reagents are highly refractory so that only advanced treatment such as anodic oxidation can degrade them [11]. The waste liquid after titrimetric analysis, which contains harmful $Mn^{2+}$ and all kinds of other polluting substances, needs to be discharged. This will potentially lead to environmental pollution. There are some studies on developing in situ $Mn^{2+}$ detection methods with relatively advanced but expensive instruments. Terada et al. performed online $Mn^{2+}$ detection by total-reflection X-ray fluorescence (TXRF) analysis [12]. However, the sample preparation lasted over 12 h, and tens of minutes are needed for measurement. Wang et al. used a rotating ring disk electrode (RRDE) to measure the $Mn^{2+}$ concentration [13], but this technique requires a high scan rate (100 mV/s). The resolution of cyclic voltammetry at this scan rate is low, and the operation is complex. The cost of the instrument in existing studies on in-situ $Mn^{2+}$ concentration measurement is too high to be suitable for industrial application. Furthermore, the speed and performance of such in-situ measuring methods are not satisfactory. Inductively coupled plasma (ICP) and atomic absorption spectroscopy (AAS) are common methods in scientific studies for determining $Mn^{2+}$ concentration [14, 15]. In spite of their unsuitably high instrument cost with regard to practical application, they are also only suitable for measuring trace $Mn^{2+}$ concentration rather than directly measuring the extremely high $Mn^{2+}$ concentration in the electrolytic manganese industry. An ideal $Mn^{2+}$ concentration measurement method applied to industry should have characteristics as follows: First, it should be able to directly determine high $Mn^{2+}$ concentration in electrolytic industry without cumbersome pretreatment. Second, it should be real-time, simple and cheap. Third, no polluting chemical reagents should be discharged from such method. Unfortunately, no existing $Mn^{2+}$ concentration determination method in practical application or scientific studies can meet the requirements above for industrial application. To the best of our knowledge, there is no research on developing methods specially for accurately measuring the high $Mn^{2+}$ concentration in electrolytic manganese industry yet.

$MnSO_4$ is the existence form of $Mn^{2+}$ in industrial electrolytes [1]. Its solution is transparent when its concentration is low. But when $MnSO_4$ concentrations reach dozens of grams per liter, which is its concentration level in electrolytic manganese industry, the color of its solution will be obviously pink (S1 Fig in S1 File). This means that $MnSO_4$ has ultraviolet-visible (UV-vis) light absorption. Many aqueous solutions of transition metal salts are colored. Therefore, these metal ions absorb light in the UV-vis region, and their concentration can be directly determined by UV-vis absorbance with fast speed, simple procedure and no polluting reagents [16–18]. It is reasonable to deduce that the ideal method for measuring $Mn^{2+}$ concentration in industrial electrolyte can also be developed based on measuring its UV-vis absorbance.

The concentration analysis by UV-vis absorbance is based on Lambert-Beer's law [17]:

$$T = \frac{I_t}{I_0} \tag{1}$$

$$A = -lgT = lg\frac{I_0}{I_t} = \varepsilon lc \tag{2}$$

where $I_0$ and $I_t$ are the intensity of the incident and transmitted light, respectively, $T$ is the transmittance, $A$ is the absorbance, $\varepsilon$ is the absorption coefficient (L/(g•cm) or L/(mol•cm)), $l$ is the optical path length (OPL; the length of the medium that the light passes through; common unit is cm), and $c$ is the concentration of the absorbing substance (mol/L or g/L). $MnSO_4$ solution shows color only at high concentrations up to dozens of grams per liter (S1 Fig in S1 File). So, its UV-vis absorbance is very weak. However, a detectable absorbance is essential to accurate quantification. It is necessary to enhance the absorbance intensity of $MnSO_4$ solution. The correlation between concentration and absorbance is shown by Eq (2). As can be seen, there are two ways to increase the absorbance: increasing the OPL ($l$) or the apparent absorption coefficient ($\varepsilon$).

In this work, $Mn^{2+}$ concentration in electrolyte was investigated to be directly determined with cost-effective UV-vis absorbance measurement which also discharges no pollutants. The instrumental parameters and the solution conditions for determining the $Mn^{2+}$ concentration in $MnSO_4$ electrolyte were optimized. Interference from coexisting substances on $Mn^{2+}$ absorbance was subtracted by mathematical absorbance data processing. To the best of our knowledge, this is the first study quantifying $Mn^{2+}$ in electrolyte by its UV-vis absorbance.

The idea of introducing UV-vis spectrometry into on-line detection in industry was launched in this study because of its incomparable speed and simpleness. Some substances have weak UV-vis absorption bands. But their UV-vis absorption signal will become clear in industrial process streams with high concentrations. With the growing requirement for precise process control and pollutant reduction, UV-vis detection method development in industry deserves attention. In addition, instrumental parameter adjustment of UV-vis spectrometry is to some extent neglected in chemical analysis. It was analyzed theoretically in detail in this study how adequately choosing the OPL, spectral band width (SBW) will help improving measurement precision.

## Materials and methods

### Reagents and solutions

Manganese sulfate, ammonium sulfate, ferric sulfate, cobalt sulfate, and nickel sulfate were purchased from Aladdin Industrial Corp. (Shanghai, China). Copper sulfate, zinc sulfate, magnesium sulfate, calcium sulfate, sodium silicate, arsenic acid and sulfuric acid were purchased from Sinopharm Chemical Reagent Co. Ltd. (Shanghai, China). All the reagents were of

analytical grade. The simulated electrolyte was prepared by dissolving analytical-grade manganese sulfate and other reagents in ultrapure water. Sulfuric acid was used to adjust the pH. Different solutions were prepared in 100.0 mL glass volumetric flask.

## Instrumentation

The absorption spectra were obtained using a T10CS double-beam spectrophotometer (Persee, China), wherein the OPL can be changed from 1.0 cm to 10.0 cm. The SBW of the instrument is also adjustable. Particle size was measured by a Nano Z Zetasizer (Malvern, UK). Concentrations of coexisting substances were measured by ICP (Agilent 5110, USA). Solution pH was measured by pH meter (Metler Toledo Fiveeasy, Switzerland). Each sample was measured 20 times by the spectrometer and absorbance was averaged as the final outcome.

## Results and discussion

### Determining the characteristic absorption peaks

To determine the characteristic absorption peak with which to quantify $Mn^{2+}$, we scanned a pure aqueous $MnSO_4$ solution at wavelengths from 300 to 800 nm. A $Mn^{2+}$ concentration in solution as high as 80.0 g/L was tested to ensure clear visualization of the absorption peak of $Mn^{2+}$. The SBW of the spectrometer was 2.0 nm, which is the default instrument setting. Cuvettes with OPL of 1.0 and 10.0 cm were used. A 2.0 mol/L $H_2SO_4$ solution was prepared and measured as a reference. The $SO_4^{2-}$ concentration in the reference was much higher than that in the $MnSO_4$ solution in order to find out whether the absorption spectrum of $MnSO_4$ is influenced by $SO_4^{2-}$. If $SO_4^{2-}$ also absorbs in the wavelength region studied, the spectrum will change in shape and intensity and the absorbance will be negative when $SO_4^{2-}$ concentration in the reference is much higher than that in the $MnSO_4$ solution. The spectrum of $MnSO_4$ has absorption peaks at 336 nm, 357 nm, 401 nm, 433 nm, and 530 nm, whether measured in the typical 1.0 cm cuvette or in a spectrum-enhancing 10.0 cm cuvette (Fig 1). These peaks were potentially correlated with the $Mn^{2+}$ concentration. The spectrum of $MnSO_4$ measured with ultrapure water as the reference was the same as that measured using $H_2SO_4$ as the reference. As was shown, $SO_4^{2-}$ did not affect the $Mn^{2+}$ spectrum. Ultrapure water was used as the reference in the following experiments.

### Absorption coefficient measurement and effect of pH

Solutions with different concentrations of $MnSO_4$ were prepared in 100.0 mL glass volumetric flasks, and their absorbance peaks were measured in a 1.0 cm cuvette. Fig 2 shows that the absorbance at 401 nm was not linearly related to the concentration, disobeying Lambert-Beer's law. As can be seen, when $Mn^{2+}$ concentration increases from 50.0 g/L to 100.0 g/L, the absorbance at 401nm was almost the same. The tested $MnSO_4$ solution was stored for one day. The next day, the originally transparent pink solution was brown with flocs and sediment in it (S1 Fig in S1 File).

$MnSO_4$ is unstable under neutral to alkaline conditions [19, 20]. It will reciprocally react with $OH^-$ to generate $Mn(OH)_2$ and further react with oxygen to generate insoluble oxides. S2 Fig in S1 File shows the spectra of $MnSO_4$ solutions with the same concentration but different pH values. As the pH increased, the absorbance obviously increased, especially at shorter wavelengths. This is similar to the experimental results in other articles studying the influence of scattering on spectra for turbidity measurement [21]. Turbidity from generation and oxidation of $Mn(OH)_2$ caused scattering of light in $MnSO_4$ solution, which contributes to part of

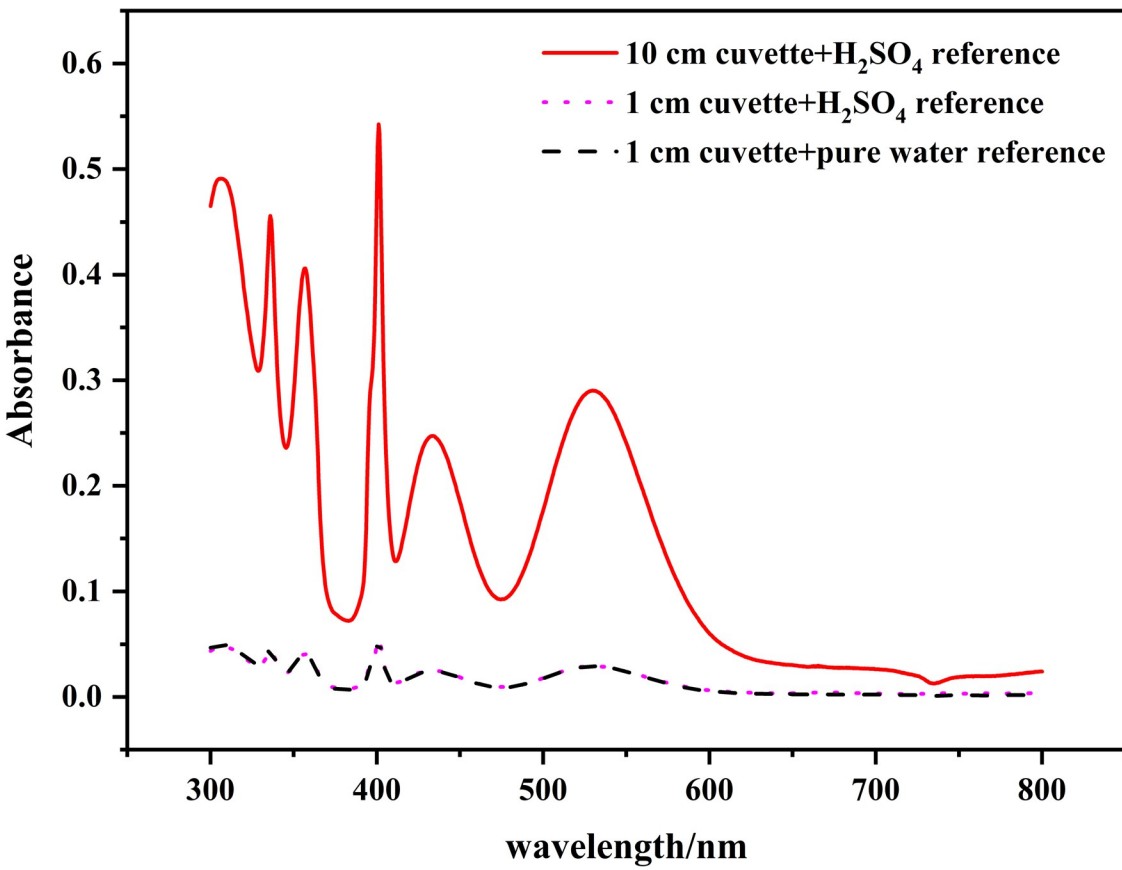

**Fig 1. Spectra of a MnSO$_4$ solution measured with different cuvettes and references.**

absorbance. As absorbance was not totally from Mn$^{2+}$ absorption, it was not linearly related to Mn$^{2+}$ concentration.

The particle size distribution in two solutions of MnSO$_4$ with pH = 1.8 and pH = 7.4 was measured, and the results in S3 Fig in S1 File show that the majority of the particles' size in the sample with pH = 7.4 were larger than those in the other solution. Therefore, scattering will be severer at higher pH which is in accordance with results in S2 Fig in S1 File.

In actual production, reductants such as SeO$_2$ and SO$_2$ are added to the electrolyte to prevent the generation of such oxidized solids [1]. However, to minimize influence from Mn (OH)$_2$ generation and oxidation, the pH should be adjusted to acidic before measurement. New solutions of MnSO$_4$ at different concentrations were prepared (10.0, 30.0, 60.0, 80.0 and 100.0 g/L) in 100.0 mL glass volumetric flasks by dissolving MnSO$_4$ powder in ultrapure water and 5.0 mL concentrated sulfuric acid (18.4 mol/L H$_2$SO$_4$) was added to each flask before dilution to volume to ensure acidic pH. Fig 3 shows the absorption spectra of different solutions. There was no obvious increase in absorbance at shorter wavelengths. Therefore, the influence from turbidity was significantly reduced. It could be seen that the MnSO$_4$ solution became stable under acidic conditions. The values of the peak absorbance of these solutions (1.0 cm cuvette), shown in S4 Fig in S1 File, were linearly related (R$^2$>0.98) to the solution concentration after pH adjustment. Fitting formulas including absorption coefficient of different peaks are also shown in S4 Fig in S1 File. The absorption peak at 401 nm was highest and was selected as the characteristic peak for the following experiments.

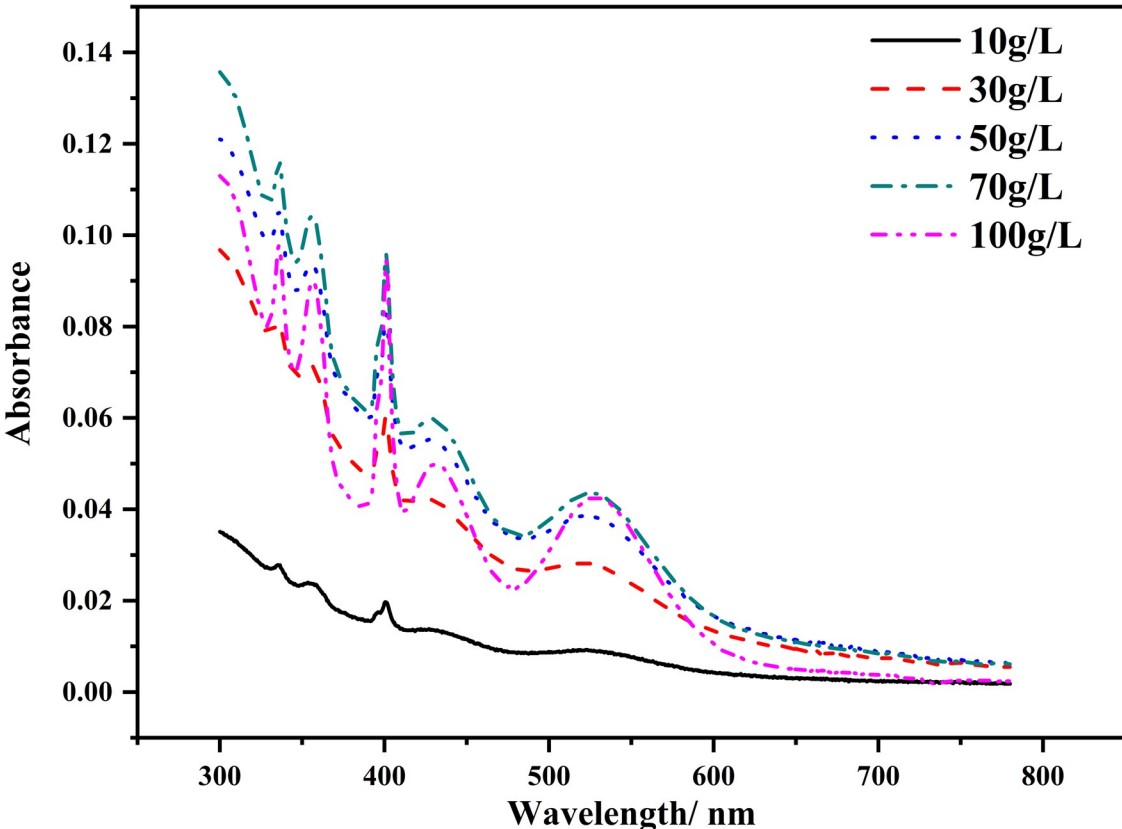

**Fig 2. Spectra of MnSO$_4$ solutions of different concentrations without pH adjustment.**

The precise pH that ensures solution stability should be determined to minimize the amount of sulfuric acid added for cost considerations. Four solutions with the same concentration but different pH values were prepared and placed in an indoor environment for 30 min before measuring their spectra (S5 Fig in S1 File). An increase in absorbance at shorter wavelengths was still obvious when pH = 3.84, as shown in S5 Fig in S1 File. However, when the pH was under approximately 2.0, the spectra were stable. Therefore, the optimum pH was 2.0 and solutions were adjusted to pH≈2.0 in the following experiment. H$_2$SO$_4$ was used as the pH controlling reagent and no new ions were introduced during pH adjustment into the MnSO$_4$ solution which can also be treated as electrolyte. If method developed in this work is used in industry for determination of Mn$^{2+}$ concentration in electrolyte, samples after measurement can be returned directly to the production line so that there will not be waste fluid discharge.

### Theoretical analysis of optimum instrumental bandwidth

Samples with a concentration of 80.0 g/L in a 1.0 cm cuvette were scanned at different SBW. Fig 4 shows that when the spectral bandwidth varied from 0.1 nm to 3.6 nm at increments of 0.5 nm, the absorbance first increased and then decreased. The relationship between the measured and real absorbance is expressed by Eq (3):

$$A_{ms} = A_{rl} + A_{err} \tag{3}$$

where A$_{ms}$ is the measured absorbance, A$_{rl}$ is the real absorbance and A$_{err}$ is the measurement error. A$_{err}$ comes from instrument or measurement. So, it is fixed under stable measuring

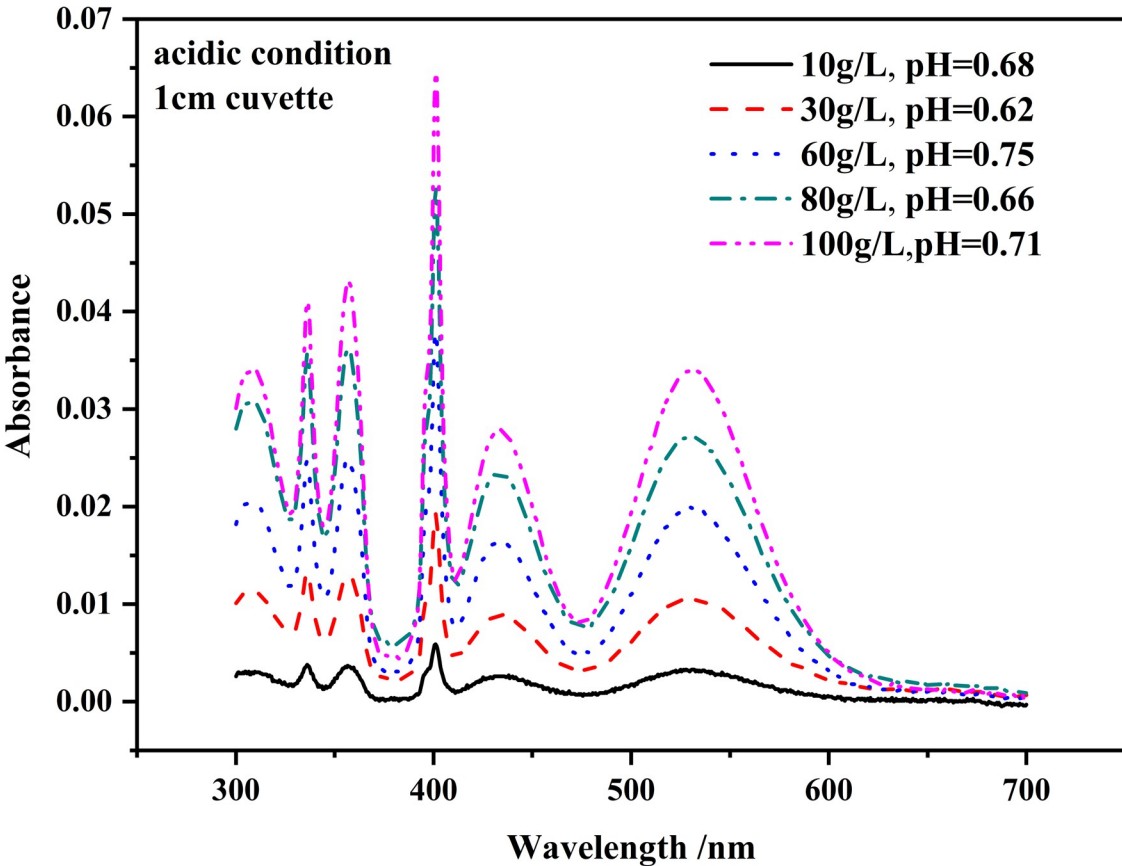

**Fig 3. Spectra of MnSO$_4$ solutions with different concentrations and low pH.**

condition. Higher $A_{rl}$ minimizes the impact of $A_{err}$ by lowering the relative error. As the absorption coefficient of MnSO$_4$ in solution is only $6.557^* 10^{-4}$ L/(g•cm) at 401 nm (SBW = 2nm, S4(c) Fig in S1 File), it is important to increase the absorbance to avoid deviation from the true value. Relationship between absorbance and SBW has been reported in previous studies [22, 23] revealing that total absorbance varies with SBW.

It is supposed that there may exist an optimum SBW producing the highest measured absorbance which will make measuring result more accurate. Fig 4 inset reveals that when the SBW changes from 0.6 to 1.6 nm at a step width of 0.1 nm, the absorbance follows the same trend as Fig 4. It first increases and then decreases before reaching a maximum when SBW is 1.0 nm. It is determined by the test results above that 1.0 nm shows to be the optimum SBW value which produces the highest absorbance. Fig 6 shows that, at an SBW of 1.0 nm, the absorbance and concentration are still linearly correlated. Therefore, the SBW was adjusted to 1.0 nm in the following experiment.

## Effect of the OPL

Eq (2) indicates that the absorbance is also affected by the OPL (*l*). Suitable path length selection is essential to the measurement of absorption spectra [24]. Measurement sensitivity is influenced by variations in the OPL. By transforming Eq (2), the relationship between the

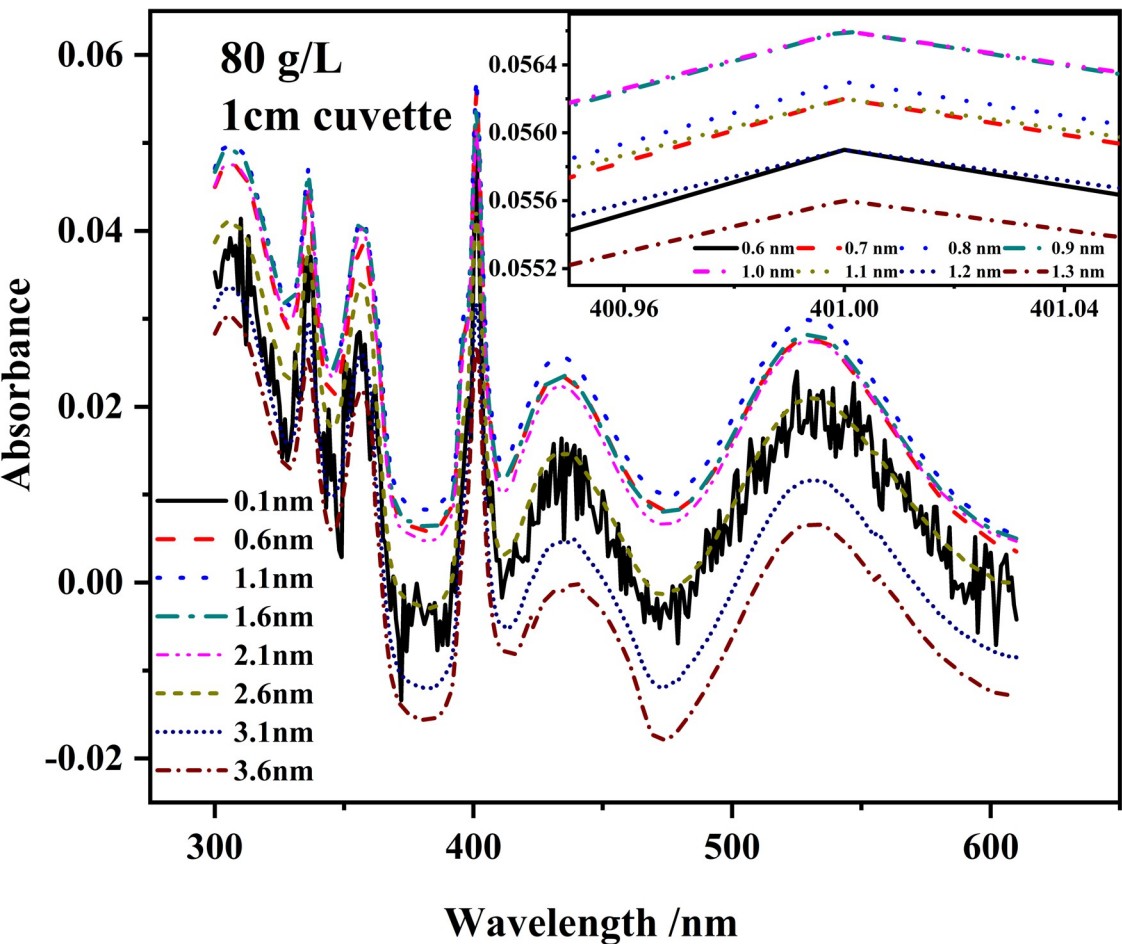

**Fig 4. Spectra of an 80.0 g/L MnSO₄ solution measured at different SBWs.** 0.1~3.6 nm, 0.5 nm increments. Inset is absorbance at approximately 401 nm at different SBWs (0.6~1.3 nm, 0.1 nm increments).

incident and transmitted light can be expressed as follows:

$$I_t = I_0 * 10^{-\varepsilon lc} \tag{4}$$

The transmitted light intensity variation per unit analyte concentration is defined as the sensitivity |SEN|, whose expression is:

$$|SEN| = |\frac{dI_t}{dc}| = ln10 * I_0 * 10^{-\varepsilon lc} * \varepsilon * l \tag{5}$$

The instrument noise of a spectrophotometer is mainly thermal noise from the detector, which affects both the incident and transmitted light [24]. Because the incident light is stronger than the transmitted light, noise from incident light can be ignored. Defining the noise from transmitted light as $\sigma_{It}$ and the deviation in concentration as $\sigma_c$, the concentration deviation caused by noise is obtained as:

$$\sigma_c = |\frac{dc}{d_{I_t}}| * \sigma_{I_t} = \sigma_{I_t}/|SEN| \tag{6}$$

A high |SEN| can lower the concentration deviation due to noise. At the same time, |SEN| is dependent on OPL as revealed by Eq (5). So, OPL may be a factor influencing measurement accuracy. S1 Table in S1 File provides a summary of the absorbance measured at the same concentration (40.0 g/L) but different OPLs (1.0, 2.0, 3.0, 5.0, 10.0 cm). Each sample was measured 20 times and the standard deviation for each sample was approximately the same, indicating that the absolute value of the deviation in absorbance from the true value was nearly the same for each sample. However, when the path length was longer, the absorbance measured at the same concentration was higher. Therefore, the relative deviation in absorbance and concentration was lowest for samples measured at the longest OPL. 10.0 cm should be the optimum OPL in the available path length series and is chosen as OPL used in the following experiment.

The $Mn^{2+}$ concentration in qualified leaching solutions and electrolytes varied within a narrow range (newly prepared leachate, 36.0~40.0 g/L; electrolyte in electrolyzer, 15.0~18.0 g/L) [1]. It is hard and important to report the precise real-time $Mn^{2+}$ concentration. A method for measuring the amount of $Mn^{2+}$ in electrolyte should be sensitive enough to detect the slight change in concentration. $MnSO_4$ solutions with concentrations of 16.0, 17.0, 18.0, 38.0, 39.0, and 40.0 g/L were measured in a 10.0 cm cuvette. S6 Fig in S1 File presents the spectra of these solutions. There were distinguishable differences in the absorbance values of the different spectra. The concentrations calculated with these absorbance values were nearly equal to the true concentrations.

## Eliminating influence from coexisting substances

Leaching solutions and electrolytes contain substances other than $Mn^{2+}$ due to additive addition and incomplete impurity removal [1]. We sampled real qualified electrolyte from one electrolytic manganese plant in China. Concentrations of all the coexisting substances were measured by ICP or acquired from the operation parameters. The data are summarized in S2 Table in S1 File. The spectrum of a mixture containing 40.0 g/L $Mn^{2+}$ and all the other coexisting substances (concentrations in accordance with data in S2 Table in S1 File) is shown in S7 Fig in S1 File. The absorbance at 401 nm of this mixture was higher than that of a pure $MnSO_4$ solution containing 40.0 g/L $Mn^{2+}$, revealing that coexisting substances will interfere with the $Mn^{2+}$ concentration measurement by increasing the absorbance.

Fig 5A presents the spectrum of a mixture solution containing all the coexisting substances except $Mn^{2+}$ at the same concentration shown in S2 Table in S1 File. The absorbance at 401 nm was nonzero. Therefore, the absorbance due to coexisting substances should be subtracted. Fig 5B was obtained by the second derivation of the spectrum in Fig 5A. The derivative value around approximately 401 nm was almost zero, which means that the absorbance around approximately 401 nm caused by coexisting substances follows a function whose order is no more than 1. Therefore, the absorbance around 401 nm from coexisting substances follow a linear functional relationship with wavelength. Thus, this absorbance value can be calculated by linear interpolation using the absorbance values at two wavelengths of absorption valleys on the left and right sides of 401 nm. In the pure $MnSO_4$ spectra, the two absorption valleys nearest to 401 nm were at approximately 386 nm and 411 nm. Therefore, A line is drawn between the two valley points. The value on this line at the wavelength of 401 nm is to be subtracted from the measured absorbance at 401 nm on the spectra of the electrolyte containing all kinds of coexisting substances. The absorbance value to be subtracted is defined as:

$$A_{sub} = A_{386} + \frac{401 - 386}{411 - 386} * (A_{411} - A_{386}) \qquad (7)$$

where $A_{sub}$ is the absorbance to be subtracted from total absorbance at 401 nm, $A_{386}$ is the

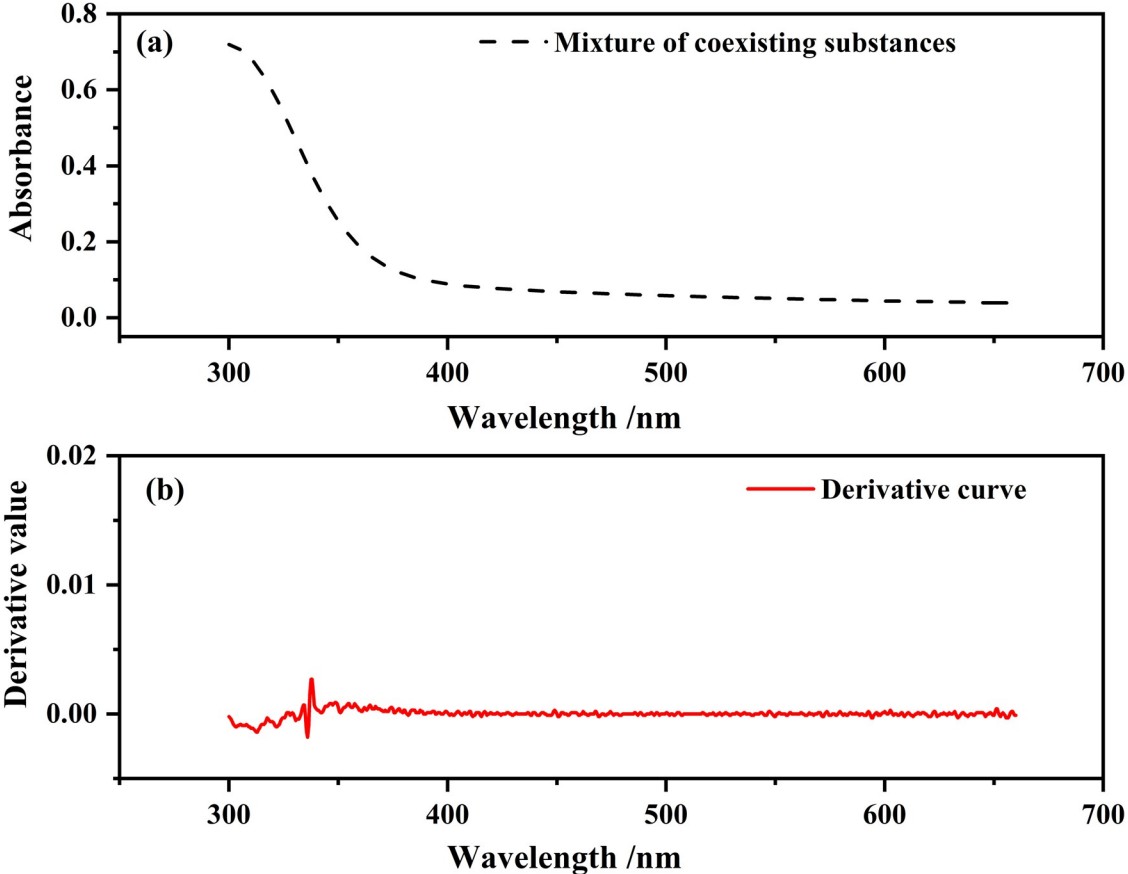

**Fig 5. Spectra of mixture solution containing all substances at their respective concentrations in industry.** (a) original spectrum and (b) second derivative spectrum.

absorbance on the spectra of the electrolyte at 386 nm, and $A_{411}$ is the absorbance on the spectra of the electrolyte at 411 nm. 386, 401 and 411 represents the corresponding wavelengths. The fraction formula in Eq (7) represents the ratio of distances between 401 nm, 411 nm and 386 nm which is also the ratio of $(A_{sub}-A_{386})$ to $(A_{411}-A_{386})$. Both coexisting substances and $Mn^{2+}$ contribute to absorbances in Eq (7) which makes $A_{sub}$ include absorbance from both coexisting substances and $Mn^{2+}$ spectrum. Therefore, the subtraction may change relationship between $Mn^{2+}$ absorbance and concentration. To test whether subtracting $A_{sub}$ from absorbance will influence the $Mn^{2+}$ concentration measurement, $A_{sub}$ were calculated with spectra of pure $MnSO_4$ solutions at different concentrations and were subtracted from the original absorbances at 401 nm to obtain the revised absorbances, which still had a good linear relationship with the $Mn^{2+}$ concentration (Fig 6). For example, $A_{386}$ and $A_{411}$ of 40.0 g/L pure $Mn^{2+}$ solution is 0.0615 and 0.085 respectively as is shown in Fig 6. Therefore, $A_{sub}$ for this solution is 0.07560 according to Eq (7). As the measured absorbance at 401 nm of the solution is 0.29925, revise absorbance at 401 nm is 0.22365 by subtracting $A_{sub}$ from measured absorbance. Revised absorbance of different solutions is all calculated in this way. The correlation coefficient ($R^2>0.999$, inset graph of Fig 6) between the revised absorbance and concentration was even higher than that between the unrevised absorbance and concentration in S4(c) Fig in S1 File. Therefore, subtracting $A_{sub}$ will not change the linear relationship between $Mn^{2+}$ concentration and absorbance. As can be seen in Fig 6, there is good linear relationship between

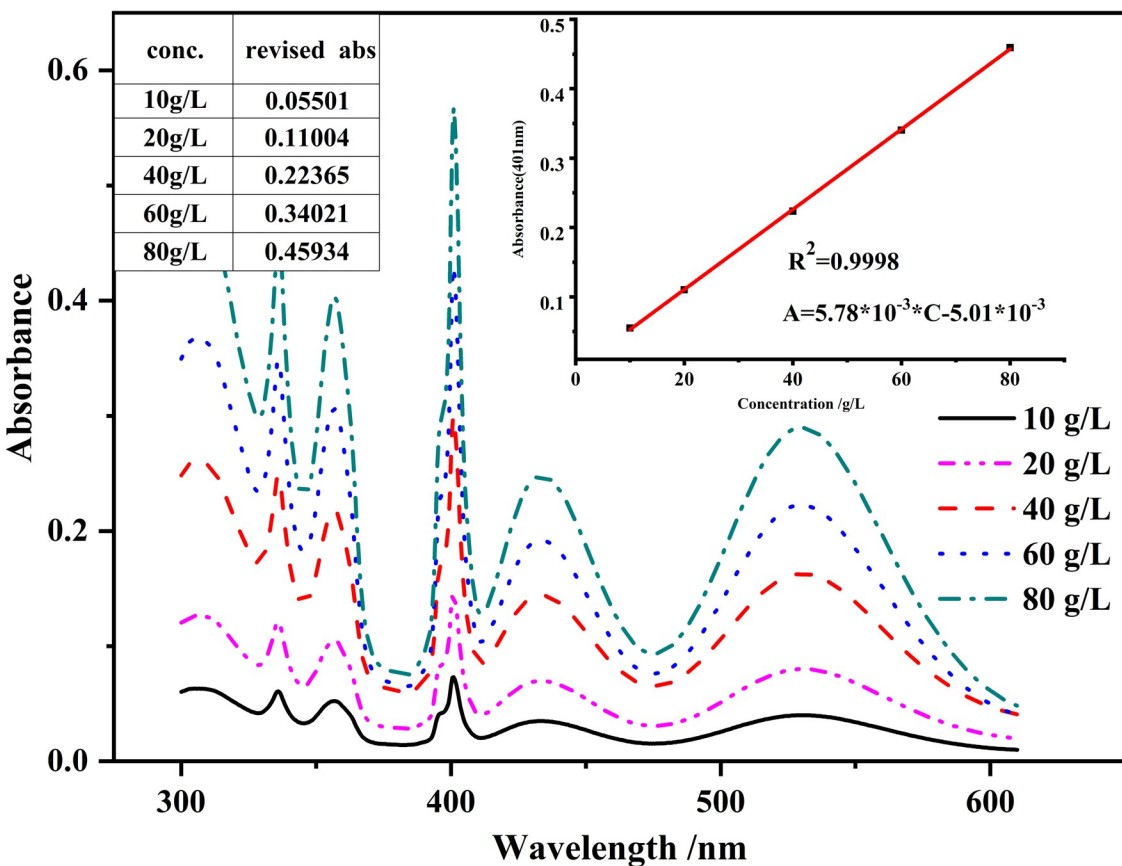

**Fig 6. Spectra of MnSO₄ solutions with different concentrations and the revised absorbance at 401nm.** Inset is the revised absorbance (by subtracting $A_{sub}$ from linear interpolation) versus the concentration.

the revised absorbance and concentration at a concentration range from 10.0 g/L to 80.0 g/L. This concentration range covers the range of $Mn^{2+}$ concentration in the electrolytic manganese production which is mentioned in the introduction section (newly prepared leachate, 36.0~40.0 g/L; electrolyte in electrolyzer, 15.0~18.0 g/L). So, the concentration range of $Mn^{2+}$ to be analyzed in the practical production is well within the linear range of the measurement method in this study. The limit of detection (LOD) and the limit of quantification (LOQ) based on absorbance of this method were calculated as 3 times and 10 times the absorbance noise of the spectrometer respectively. The absorbance noise was provided by the manufacturer as 0.0005. So, the LOD and LQD are 0.0015 and 0.005 respectively. Converted to concentration using linear equation in Fig 6, the LOD and LQD of this method are 1.1 g/L and 1.7 g/L respectively.

The method above was applied to calculate the revised absorbance at 401nm of simulated electrolyte samples containing different concentrations of $Mn^{2+}$ and all the coexisting substances at their respective concentrations in industry (S2 Table in S1 File) first. Then, the $Mn^{2+}$ concentrations were back calculated with the revised absorbance and the fitted curve in Fig 6 inset. Concentrations calculated with the unrevised absorbances and fitted curves in S4 Fig in S1 File are compared with the concentrations calculated with the revised values by method in this section (S3 Table in S1 File). It is revealed in S3 Table in S1 File that revision leads to accurate calculated concentrations with relative error less than 1.7% while the concentrations calculated with unrevised absorbances exhibit relative errors over 23.0%. Linear interpolation is

an effective method to eliminate interferences from coexisting substances and acquire accurate concentrations when other parameters, such as the pH, SBW and OPL, have been optimized.

## Conclusion

Simple UV-vis spectrometry was used to determine the $Mn^{2+}$ concentration in this study, and the developed method met the requirements for both speed and accuracy without a pollution risk. The method is easy to realize the on-line mode when the sampling procedure is automated due to the fast speed of UV-vis measurement, which further facilitates the application effectiveness. Considerable expenses, such as reagent and labor costs, are incurred for $Mn^{2+}$ detection in industry, which also generates pollution because current standard methods measuring $Mn^{2+}$ are based on titration which consumes many polluting reagents while the detection speed is low [25–27]. Application of the method developed in this work will bring about economic and environmental benefits, especially in China which is the largest electrolytic manganese producer in the world.

In this work, influence from parameters such as pH, SBW and OPL on precise $Mn^{2+}$ concentration determination was described in detail. It is stressed in this study that appropriate instrumental parameter adjustment will improve analytical performance which also should be paid enough attention to during all kinds of UV-vis measurement.

Metals such as Cu, Ni, and Co are electrolytically produced in industry, and their sulfates, which are the major constituents in their respective electrolytes, also absorb light in the UV-vis spectral region in aqueous solution [16]. Therefore, the developed method for $Mn^{2+}$ quantification, including the pH, SBW, OPL optimization and the interference removal via mathematical treatment of the absorbance, may be universally applicable. Electrolytic metal industries other than electrolytic manganese can refer to this work to develop green real-time detection methods and cut back on the contaminants discharged into environment.

## Supporting information

**S1 File.**
(DOCX)

## Acknowledgments

We thank Wenjun Zhang (WZ) for his help with instrument operation and M. Nellie for her help in language polishing.

## Author Contributions

**Conceptualization:** Zhehua Xue, Lei Li.

**Data curation:** Zhehua Xue, Lei Li.

**Investigation:** Zhehua Xue.

**Methodology:** Zhehua Xue.

**Project administration:** Lei Li.

**Supervision:** Lei Li.

**Validation:** Zhehua Xue, Lei Li.

**Writing – original draft:** Zhehua Xue.

**Writing – review & editing:** Lei Li.

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
