## [Decision Letter · Decision Letter 0]

24 Jan 2022

PONE-D-21-38889Rapid and green detection of manganese in electrolytes by UV-vis spectrometry to control pollutant dischargePLOS ONE

Dear Dr. Li,

Thank you for submitting your manuscript to PLOS ONE. After careful consideration, we feel that it has merit but does not fully meet PLOS ONE’s publication criteria as it currently stands. Therefore, we invite you to submit a revised version of the manuscript that addresses the points raised during the review process.

We look forward to receiving your revised manuscript.

Kind regards,

MARIA LUISA ASTOLFI, Ph.D.

Academic Editor

PLOS ONE

Journal Requirements:

Reviewers' comments:

Reviewer's Responses to Questions

**Comments to the Author**

1. Is the manuscript technically sound, and do the data support the conclusions?

Reviewer #1: Yes

Reviewer #2: Partly

2. Has the statistical analysis been performed appropriately and rigorously? 

Reviewer #1: Yes

Reviewer #2: N/A

3. Have the authors made all data underlying the findings in their manuscript fully available?

Reviewer #1: Yes

Reviewer #2: No

4. Is the manuscript presented in an intelligible fashion and written in standard English?

Reviewer #1: Yes

Reviewer #2: No

5. Review Comments to the Author

Reviewer #1: In electrolytic manganese industry, controlling manganese (Mn2+) concentration is very important to product quality. However, the traditional method of detecting manganese (Mn2+) concentration is complicated and time-consuming. Xue et al. reports a new method for rapid and environmentally friendly determination of Mn2+ in electrolyte and explores the best determination conditions. The following comments need to be addressed before the work can be further considered for publication.

1. The abbreviations that first appear in this article should be defined.

2. The resolution of illustrations in this paper is too low, so it is recommended to improve. There are shadows in the picture.

3. The best reading range of spectrophotometer is 0.2-0.8. The absorbance values of some illustrations in this paper are not in this range. Does it have any influence?

4. The linear range and detection limit of this method?

5. According to Fig S7, coexisting substances have certain influence on the detection of Mn2+. Can other methods be used to reduce the influence?

6. There are some formatting errors in the references, please check them carefully.

7. More recent work is suggested to be referred: Electrochem. Commun. 2021, 123, 106912; Anal. Chem. 2017, 89, 2191-2195.

Reviewer #2: 1) The continuity in the text is missing in the starting paragraph and appears as fragmented sentences.

2) line 61- current methods [8,9] - the term "current methods" may be modified for readability. The unique results observed in the cited reference should be properly highlighted.

3) line 75- expensive instrument like.... so on [12,13]- avoid short phrases such as so on, rather briefly present different instruments and discuss their advantages and disadvantages.

4) line 75- few researchers .. the author must cite with proper references and present the supporting data.

5) Uv-visible absorption is highly sensitive detection method and it is prone to possible interferences from other components since many substances absorb broad regions of the spectrum. Author must run liquid chromatography of the sample to rule out any interferences.

6) Author claimed that the Uv-visible approach for quantitative analysis adopted in current studies to be accurate without pollution risk, but H2SO4 used in this study to reduce turbidity of sample is highly corrosive! how would author explain this?

6. PLOS authors have the option to publish the peer review history of their article (what does this mean?). If published, this will include your full peer review and any attached files.

Reviewer #1: No

Reviewer #2: No

---

## [Author Response · Author response to Decision Letter 0]

9 Feb 2022

Response to Reviewers

Dear editor and reviewers,

Thank you for the editor’s email informing us of manuscript revision and the reviewers’ comments on our manuscript entitled "Rapid and green detection of manganese in electrolytes by UV-vis spectrometry to control pollutant discharge" (Manuscript Number: PONE-D-21-38889). These comments are of great value and help for improving our manuscript. 

We read up on the comments and made corresponding response carefully. The color of newly added contents in the revised manuscript are in red. Deleted contents still exist but are marked in red and with red strikethrough. All the numbers of pages, and lines in this response to reviewers are in accordance with revised manuscript with track changes. 

Our point-by-point responses to the reviewers’ comments are as follows:

Journal Requirements:

1. Please ensure that your manuscript meets PLOS ONE's style requirements, including those for file naming. The PLOS ONE style templates can be found at https://journals.plos.org/plosone/s/file?id=wjVg/PLOSOne_formatting_sample_main_body.pdf and https://journals.plos.org/plosone/s/file?id=ba62/PLOSOne_formatting_

sample_title_authors_affiliations.pdf.

Response: We have referred to the style templates carefully and revised the manuscript style to ensure that our manuscript meets PLOS ONE's style requirements, including those for file naming.

Response: We have reviewed our reference list to ensure that it is complete and correct. We added more articles in the reference of the revised manuscript according to the reviewer's comment. This is mentioned in the Response to Reviewers.

Comments to the Author

1. Is the manuscript technically sound, and do the data support the conclusions?

Reviewer #1: Yes

Reviewer #2: Partly

Response: Work underlying this manuscript such as the design of this study, the organization and analyses of data, the paper writing and the propriety and organization of language have been taken seriously. We have done our best to make sure that this manuscript is technically sound and the data support the conclusions.

2. Has the statistical analysis been performed appropriately and rigorously?

Reviewer #1: Yes

Reviewer #2: N/A

Response: We were careful and serious with the organization and analyses of data and the propriety and organization of language to make sure that the statistical analysis has been performed appropriately and rigorously.

3. Have the authors made all data underlying the findings in their manuscript fully available?

Reviewer #1: Yes

Reviewer #2: No

Response: We have checked this manuscript carefully in the first submission to make sure that all raw data underlying our findings can be found in the figures and tables of the manuscript and its Supporting Information. During the revision this time, we checked this manuscript and its Supporting Information again carefully and tried our best to find out if we missed any raw data in the manuscript and Supporting Information. Maybe S3 table of Supporting Information missed absorbance at 386 nm and 411 nm which are raw data for calculation of the revised absorbance at 401 nm in this table. We added these data into S3 table of Supporting Information. We have made it clear in the Data Availability Statement in the manuscript PDF file that all relevant data are within the manuscript and its Supporting Information files. We ensure that we have made all data underlying the findings in our manuscript fully available.

4. Is the manuscript presented in an intelligible fashion and written in standard English?

Reviewer #1: Yes

Reviewer #2: No

Response: The language of this manuscript was carefully checked by us and polished by professional English editing service before submission. So, we are sure that this manuscript is presented in an intelligible fashion and written in standard English.

5. Review Comments to the Author

Reviewer #1: In electrolytic manganese industry, controlling manganese (Mn2+) concentration is very important to product quality. However, the traditional method of detecting manganese (Mn2+) concentration is complicated and time-consuming. Xue et al. reports a new method for rapid and environmentally friendly determination of Mn2+ in electrolyte and explores the best determination conditions. The following comments need to be addressed before the work can be further considered for publication.

1. The abbreviations that first appear in this article should be defined.

Response: Thank you for your helpful comment. We checked the manuscript again carefully to add the full name of every abbreviation to the position of its first appearance in the manuscript.

2. The resolution of illustrations in this paper is too low, so it is recommended to improve. There are shadows in the picture.

Response: Thank you for your helpful comment. We improved the resolution of all illustrations in this manuscript and uploaded them together with the revised manuscript again. Our manuscript is converted to PDF for you to review. The resolution of the illustrations indeed decreases if directly read in the PDF manuscript. There is download link for every uploaded illustration in the page where this illustration exists in the PDF manuscript. You can download the illustration to read it in a high-resolution form.

3. The best reading range of spectrophotometer is 0.2-0.8. The absorbance values of some illustrations in this paper are not in this range. Does it have any influence?

Response: Thank you for your helpful comment. The conclusion that the best reading range of spectrophotometers is 0.2-0.8 is classical but a little out-of-date due to the fast improvement of UV-vis spectrophotometers nowadays. The best reading range of a spectrophotometer is decided by its quality and performance. The spectrophotometer used in this study is a T10CS double-beam spectrophotometer (Persee, China). It is relatively expensive and has very excellent performance among all kinds of spectrophotometers from different brands. The detailed performance parameters of T10CS can be found in the manufacturer's official website.

The equation with which the best reading range is calculated can be expressed as follows:

E_r=dT/TlnT*100%

Where Er is the relative error in UV-vis measurement. T is the true transmittance. dT is the absolute measurement error of transmittance. The detailed derivation procedure of this equation can be seen in the journal article in brackets (Analytical Chemistry 1955, 27, 5, 716–725). dT is strongly associated with the quality and performance of UV-vis spectrophotometers. Poorer spectrophotometers have larger dT values. Better spectrophotometers have smaller dT values but the price will be higher. The corresponding values of Er, dT, T and absorbance are listed in the table as follows:

T % 95 94 93 92 91 90 80 70 65

Absorbance 0.022 0.027 0.032 0.036 0.041 0.046 0.097 0.155 0.187 

Er %(dT=1.0%) 20.522 17.193 14.817 13.036 11.652 10.546 5.602 4.005 3.571 

Er %(dT=0.5%) 10.261 8.597 7.408 6.518 5.826 5.273 2.801 2.003 1.786 

Er %(dT=0.3%) 6.157 5.158 4.445 3.911 3.496 3.164 1.681 1.202 1.071 

Er %(dT=0.15%) 3.078 2.579 2.223 1.955 1.748 1.582 0.840 0.601 0.536 

T % 60 50 40 36.8 30 20 15 10 5

Absorbance 0.222 0.301 0.398 0.434 0.523 0.699 0.824 1.000 1.300 

Er %(dT=1.0%) 3.263 2.885 2.728 2.718 2.769 3.107 3.514 4.343 6.676 

Er %(dT=0.5%) 1.631 1.443 1.364 1.359 1.384 1.553 1.757 2.171 3.338 

Er %(dT=0.3%) 0.979 0.866 0.819 0.815 0.831 0.932 1.054 1.303 2.003 

Er %(dT=0.15%) 0.489 0.433 0.409 0.408 0.415 0.466 0.527 0.651 1.001 

As can be seen, Er at the same T values changes with dT. If the ideal Er is set as less than around 3.5%, it can be revealed from the table above that the best T should vary in a range of around 65% to 15% at the poor dT of 1.0%. Converted from transmittance to absorbance, this range equals to about 0.2~0.8. Calculation above is the origin of the conclusion mentioned by you that the best reading range of spectrophotometers is 0.2-0.8. But the quality and performance of UV-vis spectrophotometers are improving fast nowadays. For the T10CS spectrophotometer used in this study, the nominal dT from its manual is 0.3% which is much better. The technicist from the manufacturer told us that the actual dT of this spectrophotometer can be less than 0.15%. Under such excellent dT values, if the ideal Er is still set as less than around 3.5%, the corresponding best absorbance reading range of T10CS spectrophotometer should be 0.04~>1.3 (dT=0.3%) and <0.02~>1.3 (dT=0.15%) respectively. So, the absorbance values in this paper are all within the best absorbance reading range of UV-vis spectrophotometer due to the high quality and performance of T10CS spectrophotometer.

4. The linear range and detection limit of this method?

Response: Thank you for your helpful comment. The linear range and detection limit of this method were described and calculated in the revised manuscript with track changes (Pages 19-20; lines 404-416).

5. According to Fig S7, coexisting substances have certain influence on the detection of Mn2+. Can other methods be used to reduce the influence?

Response: Thank you for your valuable comment. There is a section named "Eliminating influence from coexisting substances" in this manuscript which specially discussed the method to reduce the influence from coexisting substances on the actual UV-vis absorbance of Mn2+. Linear interpolation was found to be a good method to determine the interfering absorbance value which is to be subtracted from the measured absorbance of the electrolyte containing all kinds of coexisting substances. After subtraction, there was still linear relationship between the revised absorbance and concentration with which the Mn2+ concentration in electrolyte could be accurately calculated and the interference from coexisting substances on concentration determination was removed (see Supporting Information S3 Table). Detailed contents of this section can be seen in pages 17-20; lines 351-433 in the revised manuscript with track changes.

6. There are some formatting errors in the references, please check them carefully.

Response: Thank you for your valuable comment. We checked the reference carefully again to ensure that all formatting errors have been corrected.

7. More recent work is suggested to be referred: Electrochem. Commun. 2021, 123, 106912; Anal. Chem. 2017, 89, 2191-2195.

Response: Thanks for your helpful comment. The two journal articles mentioned in this comment were all added to the reference of the revised manuscript (reference NO. 8 and 11).

Thank you very much for your comments and advices.

Reviewer #2: 1) The continuity in the text is missing in the starting paragraph and appears as fragmented sentences.

Response: Thank you for your valuable comment. We rewrote the introduction section according to your comment. The language and contents were reorganized to make the introduction section more fluent, concise and logical (Pages 2-10; lines 39-198 in the revised manuscript with track changes).

2) line 61- current methods [8,9] - the term "current methods" may be modified for readability. The unique results observed in the cited reference should be properly highlighted.

Response: Thank you for your valuable comment. The whole sentences have been revised according to your comment (Page 5; lines 101-109 in the revised manuscript with track changes). The two cited references all used standard titrimetric method to determine Mn2+ concentration. This is the reason we cite them. Their unique results are not relevant to analytical methods development so that they are not necessary to be highlighted.

3) line 75- expensive instrument like.... so on [12,13]- avoid short phrases such as so on, rather briefly present different instruments and discuss their advantages and disadvantages.

Response: Thank you for your valuable comment. The whole sentences have been revised according to your comment (Page 6; lines 118-123 in the revised manuscript with track changes).

4) line 75- few researchers .. the author must cite with proper references and present the supporting data.

Response: Thank you for your helpful comment. Actually, after a careful search, we found no research paper on developing methods specially for accurately measuring the high Mn2+ concentration in electrolytic manganese industry. There is no reference for us to cite. We used the word "few" in this sentence of the unrevised manuscript to express negative meanings and for sake of prudent expression. This sentence was revised to avoid misunderstanding (Page 6; lines 130-132 in the revised manuscript with track changes).

5) UV-visible absorption is highly sensitive detection method and it is prone to possible interferences from other components since many substances absorb broad regions of the spectrum. Author must run liquid chromatography of the sample to rule out any interferences.

Response: Thank you for your valuable comment. The research object of this study is electrolyte in electrolytic manganese industry. It is actually an inorganic salt solution. Liquid chromatography is used to separate organic substances. So, it will not be useful for ruling out any interference in this study. Deep purification is necessary for the preparation of manganese electrolyte. So, manganese electrolyte is actually a relatively simple mixed solution containing mainly MnSO4 and some inorganic additives and trace inorganic impurities. The concentration and composition of additives and impurities in manganese electrolyte are definite and well-known which are listed in S2 Table of Supporting Information. The additives and impurities indeed interfered with the accurate UV-vis absorbance measurement of Mn2+. So, there is a section named "Eliminating influence from coexisting substances" in this manuscript which specially discussed the method to reduce the influence from coexisting substances on the actual UV-vis absorbance of Mn2+ (Pages 17-20; lines 351-433 in the revised manuscript with track changes). Linear interpolation was found to be a good method to determine the interfering absorbance values from coexisting substances which is to be subtracted from the measured absorbance of the electrolyte in this study. After subtraction, there was still linear relationship between the revised absorbance and concentration with which the Mn2+ concentration in electrolyte could be accurately calculated and the interference from coexisting substances on concentration determination was removed (see Supporting Information S3 Table).

6) Author claimed that the UV-visible approach for quantitative analysis adopted in current studies to be accurate without pollution risk, but H2SO4 used in this study to reduce turbidity of sample is highly corrosive! how would author explain this?

Response: Thanks for your helpful comment. We have taken the environmental pollution risk of discharging any waste liquid after measurement into consideration. In this study, only H2SO4 were added into the electrolyte sample to be analyzed which mainly contained MnSO4. No other ions were introduced into the electrolyte sample to be analyzed during Mn2+ concentration measurement with UV-vis approach in this study. So, waste liquid produced after being analyzed by the method in this study can be directly returned to the production line of electrolytic manganese industry. Hence there will be no discharge of waste liquid and no pollution risk for the Mn2+ concentration measurement method in this study. There is no need to worry about the corrosive H2SO4 being discharged. The contents above describing the pollution-free nature of the method in this study have already been briefly included in the unrevised and revised manuscript (Page 14; lines 287-291 in the revised manuscript with track changes).

6. PLOS authors have the option to publish the peer review history of their article. If published, this will include your full peer review and any attached files.

Response: We agree to publish the peer review history of our article.

---

## [Decision Letter · Decision Letter 1]

14 Feb 2022

Rapid and green detection of manganese in electrolytes by ultraviolet-visible spectrometry to control pollutant discharge

PONE-D-21-38889R1

Dear Dr. Li,

We’re pleased to inform you that your manuscript has been judged scientifically suitable for publication and will be formally accepted for publication once it meets all outstanding technical requirements.

Kind regards,

MARIA LUISA ASTOLFI, Ph.D.

Academic Editor

PLOS ONE

Additional Editor Comments (optional):

Reviewers' comments:

Reviewer's Responses to Questions

**Comments to the Author**

1. If the authors have adequately addressed your comments raised in a previous round of review and you feel that this manuscript is now acceptable for publication, you may indicate that here to bypass the “Comments to the Author” section, enter your conflict of interest statement in the “Confidential to Editor” section, and submit your "Accept" recommendation.

Reviewer #1: All comments have been addressed

Reviewer #2: All comments have been addressed

2. Is the manuscript technically sound, and do the data support the conclusions?

Reviewer #1: Yes

Reviewer #2: Yes

3. Has the statistical analysis been performed appropriately and rigorously? 

Reviewer #1: Yes

Reviewer #2: Yes

4. Have the authors made all data underlying the findings in their manuscript fully available?

Reviewer #1: Yes

Reviewer #2: Yes

5. Is the manuscript presented in an intelligible fashion and written in standard English?

Reviewer #1: Yes

Reviewer #2: Yes

6. Review Comments to the Author

Reviewer #1: The authors have addressed all issues reaised by the reviewers. I therefore suggest that the revised paper can be accepted for publication in its current form.

Reviewer #2: The revised manuscript is up to the mark of the journal standard. The author has answered all the review questions. The manuscript in its present form can be accepted for publication.

7. PLOS authors have the option to publish the peer review history of their article (what does this mean?). If published, this will include your full peer review and any attached files.

Reviewer #1: No

Reviewer #2: No

---

## [Editor Report · Acceptance letter]

18 Feb 2022

PONE-D-21-38889R1 

Rapid and green detection of manganese in electrolytes by ultraviolet-visible spectrometry to control pollutant discharge 

Dear Dr. Li:

I'm pleased to inform you that your manuscript has been deemed suitable for publication in PLOS ONE. Congratulations! Your manuscript is now with our production department. 

Kind regards, 

on behalf of

Dr. MARIA LUISA ASTOLFI 

Academic Editor

PLOS ONE